# Production of Bioferments from Artichoke and Asparagus Waste with High Unicellular Protein and Carotenoid Content Using *R. mucilaginosa*

Magaly De La Cruz-Noriega [1],*, Santiago M. Benites [1], Segundo Rojas-Flores [1], Claudio Quiñones-Cerna [2], Nicole Terrones Rodríguez [3], Heber Robles-Castillo [3], Johnny Huanes-Carranza [2] and Karol Mendoza-Villanueva [4]

[1] Vicerrectorado de Investigación, Universidad Autónoma del Perú, Lima 15842, Peru; santiago.benites@autonoma.pe (S.M.B.); segundo.rojas.89@gmail.com (S.R.-F.)
[2] Departamento de Microbiología y Parasitología, Facultad de Ciencias Biológicas, Universidad Nacional de Trujillo, Juan Pablo II Av., Trujillo 13008, Peru; cquinonesc@unitru.edu.pe (C.Q.-C.); jhuanesc@unitru.edu.pe (J.H.-C.)
[3] Laboratorio de Biotecnología e Ingeniería Genética, Facultad de Ciencias Biológicas, Universidad Nacional de Trujillo, Trujillo 13008, Peru; nterronesa.30@gmail.com (N.T.R.); hrobles@unitru.edu.pe (H.R.-C.)
[4] Investigación Formativa e Integridad Científica, Universidad César Vallejo, Trujillo 13001, Peru; mendvillan@gmail.com
* Correspondence: mdelacruzn@autonoma.edu.pe

**Abstract:** Microorganisms' degradation of agro-industrial waste produces bad odors and greenhouse gases that contribute to global warming. Consequently, eco-friendly, sustainable biotechnological alternatives to this waste are sought to provide additional value, which is why this study's objective was to develop a method of producing unicellular proteins from artichoke and asparagus agro-industrial waste using *Rhodotorula mucilaginosa* as a producer organism. Agricultural soil was collected from the Universidad Nacional de Trujillo (Peru), and *R. mucilaginosa* was isolated and identified using biochemical tests. Proteins and carotenoids were produced from artichokes and asparagus residues using the *R. mucilaginosa* yeast. Four substrate concentrations (10, 20, 30, and 40%) and a pH range (5–8.1) were used. They were incubated at 30 °C for 72 h. The results showed that protein and carotenoid yield varied according to pH and substrate concentration. Artichoke residues reached a maximum protein yield of 25.98 mg/g and carotenoids of 159.26 µg/g at pH 5–6.6, respectively. Likewise, the asparagus residue showed a maximum protein yield of 20.22 mg/g and a carotenoid yield of 358.05 µg/g at a pH of 7.1 and 6.6, respectively. This study demonstrated the potential of artichoke and asparagus agro-industrial residues for the production of unicellular proteins and carotenoids using *R. mucilaginosa*. Further, it represents an appropriate alternative to properly managing agro-industrial waste, giving it an economic value.

**Keywords:** artichoke; asparagus; bioferments; carotenoids; *Rhodotorula mucilaginosa*

## 1. Introduction

Separating and managing organic and non-organic waste highlights the need to adopt ecologically, economically, and socially acceptable strategies to address this global issue [1]. The utilization of agricultural waste to produce value-added products such as energy, biofuels, and biofertilizers is a key strategy in sustainable management and the circular economy, thereby avoiding the release of pollutants into the environment [2]. Within this context, the reuse of agro-industrial waste generated in the production of the La Libertad region, Peru, can be considered a step towards emphasizing the circular economy within the agro-industry, where waste reduction is paramount [3].

Given the significant generation of waste as the agro-export sector has become the fastest-growing economic and productive activity in recent years, with Peru being the

second-largest asparagus producer in the world after China, as well as the third agricultural product exported after coffee and cocoa. Furthermore, the global cultivation of artichokes is concentrated 15% in Peru and Argentina. Both crops represent important sources of utilization regarding waste due to their carbohydrate content in the form of cellulose, lignin, and hemicellulose. Among the generated waste, stems, leaves and vegetable pith can be used as substrates in biotechnological processes [4].

The search for microbial sources capable of fermenting waste and the optimization of process parameters are areas focused on the economic production of pigments. In this context, yeasts have been described as relevant sources of proteins, vitamins, and pigments. Certain yeasts are prolific producers of carotenoid pigments useful for the pharmaceutical, cosmetic, and food industries [5]. Several yeast strains have been studied, which are more suitable for large-scale production in fermenters than algae or molds due to their unicellular nature and high growth rate [6]. Yeast genera reported to be capable of accumulating β- and γ-carotenoids include *Rhodotorula*, *Rhodosporidium*, *Sporobolomyces*, and *Phaffia* [7]. Some species of *Rhodotorula*, such as *R. rubra* and *R. glutinis*, produce β-carotene, torulene, and torularhodin, the latter having the highest oxidation level among carotenoids [8]. What is more interesting is that various alternative carbon and nitrogen sources have been tested to replace commercial ones in biotechnological processes for obtaining carotenoid pigments [9].

Various authors have conducted studies on low-cost substrates such as glycerol waste, sugarcane, and hydrolysates of agro-industrial waste, examining their viability to meet yeast growth requirements. For example, Li et al. (2022) described molasses and sugarcane as economical substrates for producing carotenoids by *R. mucilaginosa* CCT-7688, with a maximum of 1794.2 μg/L [10]. Furthermore, the production of pigment in banana peel extract with *R. mucilaginosa* UANL-001L achieved the highest yield of 317 μg/g of carotenoids [11]. Additionally, Ghilardi et al. (2020) tested olive mill waste for fermentation by *R. mucilaginosa* in carotenoid production and reached a maximum of 7.3 mg/L, mainly in torulene and torularhodin [12]. Therefore, these studies demonstrate the feasibility of using low-cost substrates in carotenoid pigment production.

Usmani et al. (2022) conducted a review on the biotechnological valorization of sugar beet pulp waste from the sugar industry, which, through processes such as acid hydrolysis, hydrothermal techniques, and enzymatic hydrolysis, are converted into value-added products, serving as raw materials for biofuels, bio-hydrogen, biodegradable plastics, and chemicals such as lactic acid, citric acid, alcohols, microbial enzymes, single-cell proteins, and pectic oligosaccharides [13]. Meanwhile, Pathak et al. (2022) presented a study on the added value of jackfruit peel, applied in the bioremediation of water contaminated with dyes, contributing to pollution reduction [14]. The accumulation of agro-industrial waste generates greenhouse gases, hurting the environment. Therefore, it is necessary to explore alternatives to use these waste materials as raw materials for carotenoid (pigment) production, replacing synthetic ones with a sustainable and eco-friendly alternative. They also find application in the refined food and balanced animal feed industries, for example, improving the skin pigmentation of chickens and yolk.

Therefore, the present research aims to take advantage of the waste generated by the agro-industry in the La Libertad region, specifically the bracts of artichoke and asparagus peels, for the production of compounds such as carotenoids and unicellular protein using the *R. mucilaginosa* yeast. This yeast strain is considered biotechnologically useful, as it exhibits excellent adaptation to low-cost substrates and stress resistance, offering potential prospects as a future source of natural pigment production.

## 2. Materials and Methods

### 2.1. Isolation and Biochemical Identification of R. mucilaginosa

Agricultural soil was collected from the Universidad Nacional Trujillo, Peru. For the isolation, a suspension of the soil sample was prepared in sterile physiological saline solution (10 g/100 mL), through serial dilutions; from the last $10^4$ dilutions, 0.1 mL per

surface was sown in Petri dishes containing Sabouraud Agar, and incubated at 35 °C for 24 h. After incubation, the orange-pink pigmented colonies were identified, isolated, and purified as TQ21. Yeast identification was performed with enzymatic and colorimetric methods (WalkAway 96) using the MicroScan system (Minnesota City, MN, USA) [15]. TQ21 pure culture suspensions were prepared according to the Microscan kit turbidity standard and incubated aerobically at 37 °C for 4 h. The color changes indicated positive results due to the culture's enzymatic activity.

### 2.2. Conditioning and Pretreatment of Agro-Industrial Waste

Artichoke and asparagus bract waste came from agro-industrial companies in the La Libertad region (Peru). Residues with fungal damage were discarded. Agro-industrial waste was washed with sodium hypochlorite (100 ppm) for 10 min and rinsed three times with distilled water, allowing it to settle for 15–20 min. Then, it was dried at 60 °C for 6 h. The dehydrated residues were ground and sieved to a pore size of 0.45 mm. Finally, it was stored in polyethylene bags at room temperature [16] (Figure 1A). Subsequently, from 2% (*g/v*) of conditioned artichoke and asparagus residues, it was subjected to bleaching with 2% sodium hypochlorite under shaking conditions for one hour at room temperature. Then, it was vacuum filtered, and each agro-industrial waste was washed with distilled water until the chlorine residues were eliminated. The blanched artichoke pomace (RAB) and the artichoke were hardened in a plastic tray and dried in an oven (MEMMERT, UF260 PLUS) at 60 °C for 12 h. Finally, it was pulverized and stored in polyethylene bags at room temperature until use (Figure 1B).

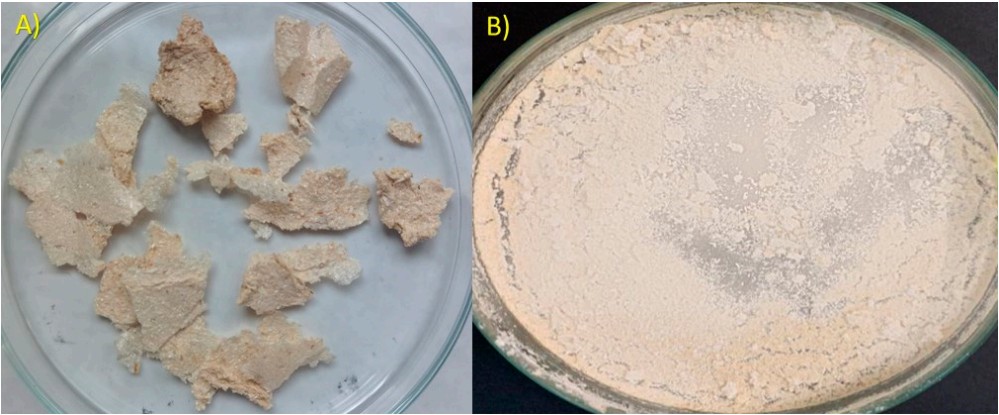

**Figure 1.** (**A**) Asparagus waste and bleached artichoke waste (**B**) as substrates for the production of bioferments using *R. mucilaginosa*.

### 2.3. Preparation of Microbial Culture

A microbial culture of *R. mucilaginosa* was enriched in 100 mL of medium containing 15 g/L yeast extract, 7.5 g/L peptone, and 0.25 g/L magnesium sulfate, which was adjusted to pH 5.0. It was incubated (MMM, ECOCELL, 111) at 3 °C at 130 revolutions per minute (rpm) for 24 h. Then, 10% of the enriched culture was inoculated into culture media prepared from pretreated agro-industrial wastes [17].

### 2.4. Fermentation of Agro-Industrial Wastes for the Production of Unicellular Protein and Carotenoids

The effect of the proportion of broth and pH on artichoke waste was evaluated for the production of unicellular protein and total carotenoids. Two milliliters of 24-h-old *Rhodotorula* sp. Culture were transferred to each of the 150 mL Erlenmeyer flasks containing 2 g of sterile waste and 20 mL of glucose broth composed of 2% glucose, 0.5% peptone, 0.1% yeast extract, and 0.5% sodium chloride, achieving different broth proportions of 5, 10, 15, and 20 mL, and pH levels of 5, 6.6, 7.1, and 8.1 [18]. The flasks were thoroughly shaken and incubated at 30 °C for 72 h (LBSI-100A, LABNICS, Glasgow, UK). During the

incubation, the flasks were manually agitated twice a day. In the end, samples were taken to determine yeast growth, total carotenoids, and total protein.

### 2.5. Measurement of Unicellular Protein and Carotenoids

The fermented residual biomass was dried at 40 °C with forced ventilation in an oven for 24 h. The protein content of the dried residual biomass was estimated using the Bradford method [19]. The yield of total carotenoid production and cell growth was determined using UV-VIS spectrometry (Jenway, Essex, UK) [20]. The total carotenoid content of the yeast cells was calculated and expressed as the total fraction of carotenoids (*FTC*) in micrograms per gram (μg/g) and the volumetric concentration of carotenoids (*CVC*) in micrograms per liter (μg/L) [21].

$$CVC \ (\mu g/L) = \frac{(A)(V)(10^6)}{E_{1cm^{1\%}}(1000)} \tag{1}$$

$$FTC \ (\mu g/g) = \frac{CVC \left(\frac{\mu g}{L}\right)}{CBC \left(\frac{g}{L}\right)} \tag{2}$$

where *A* is the absorbance at 501 nm, *V* is the total volume of carotenoids (mL), and the extinction coefficient ($E_{1cm^{1\%}(DMSO)}$) is 2040.

### 2.6. Statistical Analysis

For the statistical analysis, the software InfoStat.2020 was used. The assumption of normality was checked, and ANOVA was applied to compare the means of the dependent variables of the process generated from 2 treatments (pH and initial concentration of the medium) at four experimental levels for each waste (artichoke, asparagus). Subsequently, to identify differences between pairs of means of the factor levels, the LSD Fisher test was used with Alpha = 0.05.

## 3. Results and Discussion

A yeast strain with coral, pink-colored pigment was isolated from soil samples as pure cultures (designated as TQ21). In biochemical tests, the TQ21 culture showed assimilation of amino acids such as hydroxyproline, leucine, tyrosine, proline, glycylglycine, glycyl-proline, arginylarginine, and seryltyrosine (Table 1). It also fermented sucrose type 2 and glucosamine and tested positive for urease. However, according to Al Turki et al. (2016) in their study on yeast isolation and characterization, *R. mucilaginosa* was identified as glucosamine-negative [22], which is consistent with the findings of El-Ziney et al. (2018). Nonetheless, variable results regarding glucosamine have been observed in species such as *Rhodotorula cycloclastica* [23,24]. Furthermore, it has been found that some yeast species exhibit metabolic variability in their ability to utilize glucosamine as a carbon source [25]. Based on the morphological and biochemical characteristics, the yeast culture was identified as *R. mucilaginosa*.

Table 2 presents the results of experimental fermentation trials using blanched artichoke waste with *Rhodotorula mucilaginosa* for the production of single-cell proteins and carotenoids. The data show different pH levels and initial culture medium ratios (mL/g). When varying the pH, it is observed that the maximum dry biomass yield (38.42%) and the highest concentration of carotenoids expressed as FTC are achieved at a pH of 6.6 (159.26 μg/g), whereas for carotenoids expressed as CVC are achieved at a pH of 8.1 (549.02 μg/L). The pH is a parameter that significantly influences both cellular biomass and pigment production. In a study conducted by Zhao et al. (2019), the highest yield was achieved at an optimal pH of 5.0; however, no significant differences in biomass and carotenoid production were reported at a pH of 6 and 7 [20]. On the other hand, the alkalinization of the culture medium is a stressful factor that leads to an over-synthesis of polysaccharides and also induces the expression of genes responsible for cellular glu-

cose metabolism [26]. This would explain the increase in the volumetric concentration of carotenoids at a pH of 8.

**Table 1.** Biochemical characterization of *R. mucilaginosa* using MicroScan.

| Rapid Yeast Test ID | Result |
|---|---|
| Hydroxyproline | + |
| Isoleucine | + |
| Proline | + |
| Tyrosine | + |
| Glycine | − |
| Glycylglycine | + |
| Glycylarginine | − |
| Glycylproline | + |
| Arginylarginine | + |
| Lysylalanine | − |
| Seriltyrosine | + |
| Urease | + |
| Indosylphosphate | − |
| Histidine | − |
| Sucrose 1 | − |
| Sucrose 2 | + |
| Glucopyranoside | − |
| Fucopyranoside | − |
| Cellobiose | − |
| Galactosan | − |
| Glucosamine | + |

Source: data from the MicroScan analysis.

**Table 2.** Production of single-cell protein and carotenoids using *Rhodotorula mucilaginosa* through fermentation of blanched artichoke waste using four different pH levels and initial culture medium ratios (means ± SD). FTC: total carotenoid fraction, CVC: volumetric concentration of carotenoids.

| Treatment | Ranges | Total Protein (mg/g Dry Residue Ferment) | FTC (µg/g) | CVC (µg/L) | Dry Biomass Yield (%) | *p*-Value |
|---|---|---|---|---|---|---|
| pH | 5 | 25.97 ± 0.68 | 71.36 ± 35.85 | 388.88 ± 148.39 | 17.86 ± 5.65 | <0.05 |
| | 6.6 | 25.37 ± 1.56 | 159.26 ± 48.58 | 431.37 ± 80.99 | 38.42 ± 16.42 | |
| | 7.1 | 24.45 ± 1.41 | 152.39 ± 59.57 | 478.76 ± 106.76 | 31.07 ± 5.61 | |
| | 8.1 | 23.55 ± 0.42 | 61.66 ± 17.22 | 549.02 ± 160.27 | 11.27 ± 0.21 | |
| Initial culture medium ratio (mL/g) | 10 | 17.02 ± 0.01 | 31.12 ± 1.15 | 552.29 ± 10.20 | 5.63 ± 0.10 | <0.05 |
| | 20 | 17.56 ± 0.42 | 46.50 ± 16.88 | 668.30 ± 118.09 | 6.82 ± 1.20 | |
| | 30 | 18.15 ± 0.40 | 43.1 ± 6.83 | 648.69 ± 51.49 | 6.62 ± 0.53 | |
| | 40 | 18.12 ± 0.34 | 47.42 ± 14.11 | 676.47 ± 104.33 | 6.90 ± 1.06 | |

The total protein production slightly decreases as the pH increases, but this decrease is not significant. On the other hand, when analyzing the tested initial culture medium ratios, it is found that a ratio of 40 mL/g results in a higher concentration of carotenoids (676.47 µg/L CVC and 47.42 µg/g FTC) and dry biomass yield (6.90%). However, total protein production is similar across all initial culture medium ratios. Furthermore, when

compared to the non-fermentation treatment, a significant increase ($p < 0.05$) in protein production, carotenoids, and dry biomass yield is observed.

The means obtained for the observed response variables (total protein production in mg/g and total carotenoids in μg/g) were compared using a one-way ANOVA test, revealing significant differences ($p$-value $< 0.05$) in the levels of the pH variable. The highest dry biomass yield (38.42%) was found at a pH of 6.6, and it does not significantly differ ($p > 0.05$) from the yield at a pH of 7.1 (31.07%). However, it is statistically different from the rest ($p < 0.05$). Nevertheless, for the other parameters, the statistical analysis showed no significance ($p > 0.05$).

Regarding the variable of the initial culture medium ratio of artichoke waste, the tests showed significance ($p < 0.05$) only in the total protein production (mg/g of dry waste ferment) when using an initial ratio of 30 mL/g of waste, which achieved a higher yield (18.15 mg/g). However, there was no difference ($p > 0.05$) compared to the yields obtained with the ratios of 40 mL/g and 20 mL/g (protein production of 18.12 and 17.56 mg/g, respectively). However, it was statistically different ($p < 0.05$) from what was obtained with the ratio of 10 mL/g (17.02 mg/g).

In this regard, previous studies have indicated that a pH value of 7 is optimal for carotenoid production, as well as for the growth of Rhodotorula strains, although this can occur in a broader biokinetic zone within a pH range of 3–7 [24]. On the other hand, Bell et al. (2016) indicated, in their research, that acidic conditions with the pH values of 6, 5, 4, and 3 increased the total carotenoid content measured in carrot juice by up to 27%, with this being attributed to greater solubility of crystallized carotenoids present in vacuoles of the plant material [27]. However, with other substrates, it is confirmed that carotenoids exhibit a pH sensitivity below 25%. Regarding the ratios of the initial culture medium, the results of this research indicate that it slightly influences the protein yield in artichoke waste ferments. In their study, Kot et al. (2019) used media with the addition of 5% and 10% glycerol (approximately 28–32 g/L) to achieve a high biomass yield of the *R. mucilaginosa* species [28]. It is also worth mentioning that aeration leads to a concentration gradient that favors the fermentation process [10]. Sharma and Ghoshal (2020), in their research, reported a higher carotenoid production of 819.23 mg/g under aerobic fermentation conditions using *R. mucilaginosa* [29].

Table 3 presents the results of experimental fermentation trials using blanched asparagus waste with Rhodotorula mucilaginosa for the production of single-cell proteins and carotenoids. The data show different pH levels and initial culture medium ratios (mL/g). When varying the pH, the maximum dry biomass yield (45.08%) is achieved at a pH of 6.6, and the highest concentration of carotenoids in the form of CVC (1143.79 μg/L) is attained at a pH of 5. Total protein production is similar across all pH levels, with slight variations between 19.64 and 20.22 mg/g. Furthermore, when analyzing the initial culture medium ratios, the ratio of 40 mL/g achieves the highest total protein production (20.03 mg/g), the highest dry biomass yield (8.90%), and a moderate concentration of carotenoids (637.25 μg/L CVC and 58.29 μg/g FTC).

The analysis of variance indicated the presence of significant differences ($p < 0.05$) among the means of the response variables (total carotenoids in μg/g and percentage of dry biomass yield) using different pH levels and initial culture medium ratios of blanched asparagus waste subjected to fermentation with a strain of *Rhodotorula mucilaginosa*. At a pH level of 6.6, the highest FTC yield (475.19 μg/g) was obtained, which significantly differs ($p < 0.05$) from the results obtained at the other pH levels evaluated. Additionally, when analyzing the dry biomass yield, it was found that the biomass obtained at a pH of 6.6 significantly differs ($p < 0.05$) from the yield obtained in the other treatments (pH: 8.1 and 5), while the dry biomass yield at a pH of 8.1 was the lowest, it also statistically differed from the rest. As for the dry biomass yield, the treatment with an initial proportion of 40 mL/g of waste had the highest yield (8.9%), which was not significantly different ($p > 0.05$) from the yield obtained with a waste proportion of 30 mL/g (6.58%). However, it was statistically different ($p < 0.05$) from the treatments with proportions of 20 mL/g and 10 mL/g of waste

(1.72% and 2.87%, respectively). In this regard, Tarangini and Mishra (2014) indicated that the maximum production of carotenoids was achieved at the end of the logarithmic phase at the expense of substrate utilization [30]. Similarly, Naghavi et al. (2014) reported a significant increase in carotenoid concentration and cell mass with an increase in the temperature from 10 to 30 °C. They also indicated a maximum production of pigments and cellular accumulation of 3.40 ± 0.226 g/L and 3.93 ± 0.003 g/L, respectively [31].

**Table 3.** Experimental trials of fermentation of blanched asparagus waste on the production of single-cell protein and carotenoids with *Rhodotorula mucilaginosa* in asparagus waste fermentation using four different pH levels and initial culture medium ratios (means ± SD). FTC: total carotenoid fraction, CVC: volumetric concentration of carotenoids.

| Treatment | Ranges | Total Protein (mg/g Dry Residue Ferment) | FTC (μg/g) | CVC (μg/L) | Dry Biomass Yield (%) | *p*-Value |
|---|---|---|---|---|---|---|
| pH | 5 | 20.10 ± 0.33 | 222.04 ± 66.74 | 1143.79 ± 487.67 | 23.03 ± 3.58 | |
| | 6.6 | 20.02 ± 0.45 | 475.19 ± 100.45 | 983.66 ± 219.79 | 45.08 ± 10.83 | <0.05 |
| | 7.1 | 20.22 ± 0.53 | 254.69 ± 19.51 | 748.37 ± 51.02 | 33.37 ± 1.99 | |
| | 8.1 | 19.64 ± 0.45 | 72.88 ± 29.78 | 1088.24 ± 184.52 | 6.25 ± 2.40 | |
| Initial culture medium ratio (mL/g) | 10 | 18.53 ± 0.19 | 21.40 ± 20.62 | 707.52 ± 115.83 | 2.87 ± 2.51 | |
| | 20 | 18.90 ± 0.1 | 13.35 ± 7.72 | 797.39 ± 46.33 | 1.72 ± 1.1 | <0.05 |
| | 30 | 19.20 ± 0.15 | 44.53 ± 8.72 | 673.2 ± 89.90 | 6.58 ± 0.45 | |
| | 40 | 20.03 ± 0.23 | 58.29 ± 27.26 | 637.25 ± 106.38 | 8.90 ± 2.83 | |

The experimental fermentation trials with bleached artichoke and asparagus wastes using *R. mucilaginosa* showed variations in the production of proteins and carotenoids depending on the pH and initial medium proportion. The results indicate that protein yield in artichoke waste ferments is only affected by the initial residue proportion, while dry biomass yield is affected by the medium's pH. Similarly, in the trials with asparagus wastes, pH and initial medium proportion significantly influence the total carotenoid fraction FTC (μg/g) and dry biomass yield. Other agricultural byproducts used in carotenoid production include coffee pulp and peel extracts as nutrient sources for biopigment production with *R. mucilaginosa* [32]. Ribeiro et al. (2019) evaluated the use of cassava wastewater for *R. glutinis* cultivation in their study, and the results showed high values of cell concentration (10.28 g/L), carotenoids (0.98 mg/L), and lipids (1.34 g/L) [33]. In similar studies using the *Candida tropicalis* yeast, the use of sugar cane bagasse hydrolysates resulted in 16.97 g/L of yeast biomass [34]. Similarly, Dias Rodrigues et al. (2019) studied the production of biopigments and cell concentration using sugar cane molasses with the *R. mucilaginosa* yeast CCT7688, and the results indicated a significant increase in carotenoid production and cell concentration with the increase in the amount of sugar cane molasses [35]. Therefore, the species *R. mucilaginosa* presents advantages over other species in carotenoid production due to its characteristics such as its unicellular form, rapid doubling times, and ability to grow on a wide range of substrates [36].

## 4. Conclusions

In conclusion, this study demonstrated that the fermentation of bleached artichoke and asparagus wastes using *R. mucilaginosa* is an effective strategy for obtaining wastes with high content of unicellular proteins and carotenoids. In the case of artichoke, a maximum dry biomass yield of 38.42% was observed at a pH of 6.6; for asparagus, a maximum dry biomass yield of 45.08% was observed at a pH of 6.6. The production of total carotenoids reached its peak at a pH of 6.6 in both processes, with 159.26 μg/g FTC in artichoke and 475.19 μg/g FTC in asparagus. The fermentation conditions of the artichoke and asparagus waste were carried out under laboratory conditions, which is a limitation of the study; hence, the challenge is to take it to a large scale to maximize the production of proteins and carotenoids, contributing to the sustainable use of the agricultural waste and the

production of added value compounds with applications in the food industry. Future research is recommended to use different agro-industrial wastes as raw materials for the production of carotenoids.

**Author Contributions:** Conceptualization, M.D.L.C.-N. and N.T.R.; methodology, C.Q.-C.; software, K.M.-V.; validation, J.H.-C. and H.R.-C.; formal analysis, K.M.-V.; investigation and resources, S.M.B.; data curation, S.R.-F.; writing—original draft preparation, C.Q.-C.; writing—review and editing, M.D.L.C.-N. and C.Q.-C.; visualization and supervision, S.R.-F.; project administration, S.R.-F.; funding acquisition, S.M.B. All authors have read and agreed to the published version of the manuscript.

**Funding:** This research was funded by Universidad Autónoma del Perú grant number N° 008-2022-VRI-UA.

**Informed Consent Statement:** Not applicable.

**Conflicts of Interest:** The authors declare no conflict of interest.

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
