# Peer review of "Production of Bioferments from Artichoke and Asparagus Waste with High Unicellular Protein and Carotenoid Content Using R. mucilaginosa"

_sustainability, doi:10.3390/su152015102_

Round 1
Reviewer 1 Report
Dear Authors,
The comments can be found inside the manuscript. In general, I consider that they should work better on the writing of the manuscript, focused on presenting a scientific article and not a thesis. In addition to improving the statistical analysis, the presentation of results (not to duplicate results in tables and figures) and the discussion.
I think it would be better to look for another journal that fits the article.
Regards

Author Response
Dear colleague, the authors appreciate the comments made to improve the manuscript.
The authors commented on each of the suggestions made:
Comments can be found within the manuscript. In general, I think they should work better on writing the manuscript, focused on presenting a scientific article and not a thesis. In addition to improving the statistical analysis, the presentation of results (do not duplicate results in tables and figures) and the discussion.
Ans. Fixed as stated. Also indicate that there is research on the subject worked in this journal.
Greetings
kind regards.

Reviewer 2 Report
Comments
1. In abstract, page 1, line no. 23, “under different pH" author should mention here the range of pH.
2. Abstract can be more structured and constructive.
3. Introduction can be strengthened by adding few more relevant studies (refer, https://doi.org/10.1016/j.biortech.2021.126580,
https://doi.org/10.3389/fnut.2022.1061098.
4. Authors should add the research gap and novelty of this study in Introduction section.
5. In introduction section, page 2, line no. 52, “hearts” ?? author should check it.
6. At page 3, line no. 134, … author mentioned the culture conditions of Rhodotorula sp. “at 35°C and 130 revolutions per minute (rpm) for 24 hours” whether this is optimized condition for the growth of organism?
7. In results and discussion section, author should write references as per instruction to authors. In these lines, 175-177, year are also cited with numbering, “However, according to Al Turki et al. (2016) [20]in 175 their study on yeast isolation and characterization, R. mucilaginosa was identified as glu-176 cosamine negative, which is consistent with the findings of El-Ziney et al. (2018) [21].
At page 5, line no. 207-210, It is also worth mentioning that aeration leads to a concentration gradient that favors the 207 fermentation process (Li et al., 2022). Sharma & Ghoshal (2020) in their research report 208 higher carotenoid production of 819.23 mg/g under aerobic fermentation conditions using 209 R. mucilaginosa [10,27]”.
At page 8, line no. 296-297, In this regard, Tarangini & Mishra (2014) demonstrated in their research that pH values 296 ranging from 5 to 9 favored biomass growth[29]. …………
Author should check pattern of writing all these references.
8. Author should use acronym after explaining it in full. Author should check throughout Ms.
9. References should be cross checked and as per journal guidelines.
10. Figure quality can be improved.
11. Authors must check typological and grammatical mistakes.
It can be improved by an English expert
Author Response
Dear colleague, the authors appreciate the comments made to improve the manuscript.
The authors commented on each of the suggestions made:
1. In short, page 1, line no. 23, "under different pH" the author should mention the pH range here.
Ans. Fixed as indicated.
2. The summary can be more structured and constructive.
Ans. Fixed as indicated.
3. The introduction can be reinforced by adding some more relevant studies (see, https://doi.org/10.1016/j.biortech.2021.126580, https://doi.org/10.3389/fnut.2022.1061098
Ans. The papers were reviewed and the information was included.
4. Authors should add the research gap and novelty of this study in the Introduction section.
Ans. Fixed as indicated.
5. In the introduction section, page 2, line no. 52, "hearts" ?? The author must verify it.
Ans. Ans. Corrected grammar in the manuscript.
6. On page 3, line no. 134, ... The author mentioned the cultivation conditions of Rhodotorula sp. "at 35 °C and 130 revolutions per minute (rpm) for 24 hours" if this is an optimized condition for organism growth?
The process is based on literature reviews until this process is optimally controlled.
7. In the results and discussion section, the author must write references according to the instructions to the authors. In these lines, 175-177, year are also cited with numbering, "However, according to Al Turki et al. (2016) [20] in their study on isolation and characterization of yeasts, 175 R. mucilaginosa was identified as glutinous 176 cosamine negative, which is consistent with the findings of El-Ziney et al (2018) [21].
Ans. On page 5, line no. 207-210, It is also worth mentioning that aeration leads to a concentration gradient that favors the fermentation process 207 (Li et al., 2022). Sharma & Ghoshal (2020) in their research report 208 increased carotenoid production of 819.23 mg/g under aerobic fermentation conditions using 209 R. mucilaginosa [10,27]".
On page 8, line no. 296-297, In this sense, Tarangini & Mishra (2014) demonstrated in their research that pH 296 values ranging from 5 to 9 favor the growth of biomass[29]. ............
The author must verify the writing pattern of all these references.
Fixed as indicated.
8. The author should use the acronym after fully explaining it. The author should check throughout Ms.
Ans. Fixed as indicated.
9. References must be checked and according to journal guidelines.
Ans. Fixed as indicated.
10. The quality of the figure can be improved.
Ans. It was removed from the manuscript
11. Authors must check for typological and grammatical errors.
Ans. Fixed as indicated.
kind regards.

Reviewer 3 Report
Discussion is missing. How do the results relate to the existing literature?
Conclusion is not developed. What are the limitations of the study? What future study avenues are opened because of this study? What kind of recommendation can the authors make to the related stakeholders?
Author Response
Dear colleague, the authors appreciate the comments made to improve the manuscript.
The authors commented on each of the suggestions made:
The discussion is missing. How do the results relate to the existing literature?
Ans. It was done as indicated
The conclusion is not developed. What are the limitations of the study? What future avenues of study are opened by this study? What type of recommendation can the authors make to related stakeholders?
Ans. The fermentation conditions of the artichoke and asparagus process were carried out at the laboratory level, being a limitation in the study, so the challenge is to take it to a large scale through a pilot plant, to maximize the production of proteins and carotenoids. thus contributing to the sustainable use of agricultural residues and the production of value-added compounds with applications in the food industry. The use of different agroindustrial residues as raw material for the production of carotenoids is recommended.
kind regards.

Reviewer 4 Report
Please add data of the morphological characterization of strain. Add comments by which you support the standing that used strain belongs to R. mucilaginosa.
Add data about the statistical significance in Tab. 2.
Reference 6 - name of journal is missed.
Author Response
Dear colleague, the authors appreciate the comments made to improve the manuscript.
The authors commented on each of the suggestions made:
Please add data on the morphological characterization of the strain. Add comments supporting the position that the strain used belongs to R. mucilaginosa.
Ans. The R. mucilaginosa strain was identified morphologically with orange-pink pigmented colonies, later isolated and purified as TQ21. Yeast identification was carried out using the MicroScan colorimetric and enzymatic system (WalkAway 96) (Ceballos-Garzón et al., 2019)
Add data on statistical significance in Tab 2.
Ans. Fixed as indicated.
Reference 6: the name of the journal is lost.
Ans. Fixed as indicated.
kind regards.

Reviewer 5 Report
Dear Authors,
I have read the manuscript entitled Bioferments from artichoke and asparagus waste with high unicellular protein and carotenoid content by R. mucilaginosa.
Your work has a very interesting topic and is very important for the economy and environmental protection of the country. The manuscript with implementation corrections is good written but has some errors! I would like to commend you for citing more than 60% of publications from the last five years in the introduction and discussion.
I have some observations:
Line 31: I suggest that you list the keywords in alphabetical order. This is not stated in the instructions for authors, but the text looks better.
Lines 34-36: Please delete the redundant text before first sentence of the manuscript (sustainability-2499682-v2).
Lines 171-220: Please add the details of the manufacturer of the thermostat for incubation, MicroScan system, and a dehydrator, shaker and UV-VIS spectrometer and respect the uniformity of the text (manufacturer, city, state).
Line 196: Figure 1. has been removed from the manuscript "sustainability-2499682-v1". I suggest that you illustrate the written text 2.2. of the materials and methods with Figure 1a and b.
Line 478: Here is the error! In the Table 2, the highest concentration of carotenoids (549.02) is recorded at pH 8.1.
Lines 486-488 and 540-542: In the header of tables 2 and 3, indicate the meaning of the abbreviations FTC and CVC.
Line 534: And here is the error! This data (45.62 mg/g) does not match the given value in the Table 3.
Line 543: In the Table 3, replace the column headings “Variable, Levels, and Levels (mg/g, Levels)” with “Treatment, Range, Total protein (mg/g Dry residue ferment)” so that it is written uniformly with the rest of the data in the header of the Table 2.
Lines 543-544: Presentation of the results of the yield of dry biomass in Таbles 2 and 3 obtained by fermentation of blanched wastes of artichoke and asparagus with R. mucilaginosa at three pH values (5, 6.6 and 7.1) is incorrect. It is not usual for two plant species to get the same results, nor the same values for the statistical error. The same values are shown in Figures 5 and 6 of the manuscript "sustainability-2499682-v1". Please check the results again and make appropriate corrections if necessary.
Kind regards,
Author Response
|Dear colleague, I hope you are in good health.
The authors have made the changes at the request of your suggestions and we have made the corresponding responses to each of your suggestions:
Line 31: I suggest you list the keywords in alphabetical order. This is not indicated in the instructions to authors, but the text looks better.
Ans. It was adapted according to what was recommended.
Lines 34-36: Please remove redundant text before the first sentence of the manuscript (sustainability-2499682-v2).
Ans. The results are first for artichoke Line 24-25 and then the results for asparagus are described Line 26-27.
Lines 171-220: Add manufacturer details for the incubation thermostat, MicroScan system, and a dehydrator, shaker, and UV-VIS spectrometer, keeping the text consistent (manufacturer, city, state).
Ans. It was adapted according to what was recommended.
Line 478: Here's the error! In Table 2, the highest concentration of carotenoids (549.02) is recorded at pH 8.1.
Justify because we mention highest carotenoid concentration (431.37 μg/l CVC and 159.26 μg/g FTC) are obtained at a pH of 6.6.
Ans. It was justified according to what was recommended.
Lines 486-488 and 540-542: In the heading of tables 2 and 3, indicate the meaning of the abbreviations FTC and CVC.
Ans. It was adapted according to what was recommended.
Line 534: And here is the error! These data (45.62 mg/g) do not coincide with the value given in Table 3.
Ans. It was adapted according to what was recommended.
Line 543: In Table 3, replace the column headings "Variable, Levels and Levels (mg/g, Levels)" with "Treatment, Range, Total Protein (mg/g Dried Residue Ferment)" so that it is written uniformly with the rest of the data in the header of Table 2.
Ans. It was adapted according to what was recommended.
Lines 543-544: The presentation of the dry biomass yield results in Tables 2 and 3 obtained by fermentation of blanched artichoke and asparagus residues with R. mucilaginosa at three pH values (5, 6.6 and 7.1) is incorrect. It is not common for two plant species to obtain the same results, nor the same values for the statistical error. The same values are shown in Figures 5 and 6 of the manuscript "sustainability-2499682-v1". Please check the results again and make appropriate corrections if necessary.
Ans. It was adapted according to what was recommended.
cordial greetings.

Round 2
Reviewer 1 Report
Dear Authors
Thanks for accepting some of the suggestions, however, in my opinion, the manuscript still lacks the necessary rigor to be considered an original article. In this context, my suggestion is that you can complement your research by considering how these wastes can contribute to agriculture, for example, by carrying out incubation experiments in the lab (soil + wastes), in pots in greenhouses, etc.
Regards
Author Response

(The authors gave the same response as above.)

Reviewer 2 Report
Accepted
Author Response
Dear colleague, I hope you are in good health.
Thank you very much for his comment.
cordial greetings.
Reviewer 5 Report
Dear Authors,
I thank you for considering all my suggestions for correcting the manuscript and implementing the appropriate corrections in the text. I believe that the manuscript has been sufficiently improved and is now ready for publication in Sustainability.
Kind regards,